# Incorporating Ecological Constraints into the Simulations of Tropical Urban Growth Boundaries: A Case Study of Sanya City on Hainan Island, China

**Nianlong Han** [1,2], **Ke Hu** [2], **Miao Yu** [2], **Peihong Jia** [2,*] **and Yiqing Zhang** [3]

1   School of Geography and Tourism, Huizhou University, Huizhou 516007, China; nlhan@hainanu.edu.cn
2   School of Public Administration, Hainan University, Haikou 570228, China; huk@hainanu.edu.cn (K.H.); yumiao@hainanu.edu.cn (M.Y.)
3   School of Resource and Environmental Science, Wuhan University, Wuhan 430072, China; sameenzhang@whu.edu.cn
*   Correspondence: jiaph@hainanu.edu.cn

**Abstract:** The rapid expansion of cities in tropical regions has triggered a series of problems such as the destruction of rare natural resources and decreases in the environmental resource carrying capacity and ecological security, which seriously threaten the sustainable development of tropical cities. In this study, the city of Sanya, Hainan, China, is taken as an example. A bottom-line ecological security pattern is constructed based on the remote sensing ecological index (RSEI) and the patch-generating land use simulation (PLUS) for urban growth boundary (UGB) delineation. The results show that Sanya has a good ecological background, but the overall ecological quality decreased from 2014 to 2018 due to the expansion of construction in hot spot areas. Under the natural growth scenario, the urban built-up area in Sanya in 2030 will be 73.81% greater than in 2018, mainly occupying a large amount of agricultural and ecological space, and urban expansion will not be effectively suppressed. Delineation of the UGBs combined with the ecological constraints can effectively protect the regional ecological security and control the urban sprawl, which is relatively consistent with the current planning. The results of this study demonstrate that the RSEI-PLUS-based UGB delineation perspective has a high scientific and applicability, and they provide a reference for the coordinated ecological–economic sustainable development of ecologically fragile cities in tropical areas.

**Keywords:** ecological constraints; PLUS; urban growth boundary; tropical city





## 1. Introduction

The rapid development of urbanization has led to the disorderly expansion of urban built-up areas and their encroachment on ecological spaces, which has resulted in a series of problems such as waste of land resources, a decrease in biodiversity, degradation of ecosystem services, and destruction of tropical natural resources, all of which seriously restrict the sustainable development of the economy and environment [1–3]. Sanya is a famous international tropical coastal tourist city located on the southernmost tip of Hainan Island, China. Driven by the development of tourism, Sanya has experienced rapid urban expansion and population growth in recent decades. In the process of urban development, a series of ecological and environmental problems have emerged, such as the destruction of precious tropical rain forests, fragile mountains, key ecologically fragile areas in the coastal zone (coral reefs), and rare freshwater ecosystems. A series of policies have been formulated by the government to protect the ecological security from the damage of urban expansion, which call for delineation the urban development boundary (UGB) to prevent urban sprawl and reserve space for future development [4]. Some cities have realized that it is important to delineate UGB to restrain the disorderly urban growth, and corresponding policies have been proposed, such as the basic ecological line in Shenzhen, China [5]. Thus,

how to scientifically guide urban development and coordinate the relationship between urban growth and the ecological environment has become an urgent problem in current research. In this context, urban boundary management policies and tools to curb urban sprawl have emerged. The urban growth boundary (UGB) concept originated in Salem, OR, USA, in the 1970s and was used to delineate rural and urban land. Subsequently, it was effectively applied in Portland, Oregon, USA [6]. The UGB concept mainly emphasizes internal growth, and it aims to ease the tension between urban development and protection of the natural environment, optimizes the urban spatial structure, and formulates scientific and reasonable control policies. This is is of practical significance for sustainable economic and social development and, therefore, is widely used in China [7].

Previous research on UGBs includes several main types. The first type is ecologically oriented, which analyzes the constraints of urban growth based on the ecological sensitivity or ecological suitability, establishes ecological safety patterns, prioritizes the no-build zone in urban space, and draws on the concept of anti-planning to delineate the UGB [8–10]. The second type involves predicting the future development pattern of a city mainly based on the current urban land use pattern, development pattern, policy guidelines, and other conditions. For example, Tayyebia et al. used neural networks, the geographic information system (GIS), and remote sensing to delineate the UGB of Tehran [11]. Harig et al. took into account the spatial development planning and legal framework in Germany and produced a UGB using building footprints and generally available topographic data as the input for three study areas (Frankfurt/Main, the Hanover region, and rural Brandenburg) [12]. Chakraborti et al. used an artificial neural network (ANN) model and a set of landscape metrics to delineate the UGB in the Siliguri Municipal Corporation [13]. In addition, geo-simulation models such as the cell automation (CA) [14,15], the CA-Markov model [16], slope, land use, exclusion, urban extent, transportation, hillshade (SLEUTH) model [17], the conversion of land use and its effects at a small regional extent (CLUE-S) model [18], the future land use simulation (FLUS) model [19], and the agent-based model [20,21] are also often used to predict the future development patterns of cities. However, these modeling approaches usually lack attention to ecological and environmental factors, and the simulated urban spatial patterns are not conducive to sustainable ecological, social, and economic development. The third type of method is a combination of the first two methods, which not only considers the development needs of the urban space but also focuses on the requirements for the protection of the ecological environment; considers the requirements for economic, social, and ecological development; and is conducive to guiding the smart growth of urban land [22–25].

The ecologically oriented method involves more ecological factors, but the factor weight setting is more subjective, and there is no unified standard, which is not conducive to objectively reflecting the actual situation of the regional ecological environment. The remote sensing ecological index (RSEI) is entirely based on the use of remote sensing technology to extract the ecological factors, so it can be used to monitor the regional ecology and quickly identify ecologically sensitive areas. It has the advantages of easy access, timeliness, and avoidance of the uncertainty caused by human intervention [26], and it has been successfully applied.

Regarding the urban growth simulation models, the CA-Markov model lacks consideration of the spatial distribution changes. The CLUE-S model calculates the influences of the driving factors on land-use changes using the logistic regression method, which only takes into account the linear relationships between the individual variables [27]. The FLUS model adds adaptive inertia coefficients and a competition mechanism, so it can simulate land-use changes more realistically; however, it has the disadvantage that it is only trained based on the land class samples from the first-phase land use data and lacks analysis of historical land-use changes [28]. The patch-generating land use simulation (PLUS) model overcomes the shortcomings of the FLUS model and provides a better explanation of the land-use change mechanism through the use of two-phase land-use data. In addition, the PLUS model combined with the CA model and the patch-generating simulation strategy is

more effective in simulating complex changes in multi-type land classes [29]. Therefore, this study proposes a RSEI-PLUS-based method to delineate the UGB.

Previous studies have focused on less urban tropical island regions, which contain many important natural resources and are indispensable to the sustainability of global ecosystems. In addition, due to the closed nature of islands and their limited resources and environmental carrying capacity, the ecology of islands faces great challenges in the context of rapid urbanization and population growth. Sanya is a famous international tropical coastal tourist city located on the southernmost tip of Hainan Island, China. Driven by the development of tourism, Sanya has experienced rapid urban expansion and population growth in recent decades. In the process of urban development, a series of ecological and environmental problems have emerged, such as damage to or destruction of precious tropical rain forests, fragile mountains, key ecologically fragile areas in the coastal zone (coral reefs), and rare freshwater ecosystems. Moreover, the ecological security and resource carrying capacity have declined, threatening the sustainable development of the city's ecology, society, and economy [30]. Therefore, how to ensure the ecological safety of this region during the process of urban development is a problem that urgently needs to be solved. There is an immediate need to coordinate the contradiction between ecological protection and urban spatial growth, and ensure the region's ecological safety and the sustainable development of cities. Delineation of UGB scientifically and reasonable, is one of ways to solve the problem, and it is also the focus of current territorial space planning in China.

In this study, Sanya was taken as an example, and the RSEI was integrated into a simulation of the UGB of this tropical city. The research goals of this study were to solve the following problems. First, we realized the spatial assessment of the ecological quality of Sanya based on the RSEI and constructed the urban ecological security pattern of Sanya based on the RSEI results. Second, based on the PLUS model, we explored the differences in future urban built-up land growth simulations under natural growth and ecological constraint scenarios and delineated the UGB on this basis. Finally, based on comparison of the delineation results and territorial spatial planning, recommendations were made for achieving sustainable urban development of Sanya. In this study, we attempted to verify how the combination of ecological constraints and geo-simulation models can be adapted to tropical cities to ensure future sustainable development while providing new technical application tools for UGB delineation in the tropics.

## 2. Materials and Methods

### 2.1. Study Area

The city of Sanya is located on the southernmost tip of Hainan Island, China (18°09′34″–18°37′27″ N, 108°56′30″–109°48′28″ E). The landscape is dominated by mountains in the north, hills in the southeast, and coastal plains in the southwest (Figure 1), and it has a tropical maritime monsoon climate. The average annual temperature is 26 °C, with a maximum temperature of 29 °C in June and a minimum temperature of 21 °C in January. The average annual precipitation is 1392.2 mm.

Sanya is bordered by the South China Sea to the south and is surrounded by mountains on the other three sides, and the terrain slopes gently from north to south. This region has rich and diverse types of ecosystems, including mountains, water, forests, cultivated land, and sea, and its natural resources are scarce, fragile, and sensitive. Since the 1980s, due to the rapid urbanization brought about by tourism development, the ecology of Sanya has been damaged to some extent, including by the encroachment of urban development and construction on and destruction of mountain and forest ecosystems, the threat posed by land reclamation to the coastal ecology, the encroachment of agricultural development on water systems and forest, and weak urban environmental infrastructure [30].

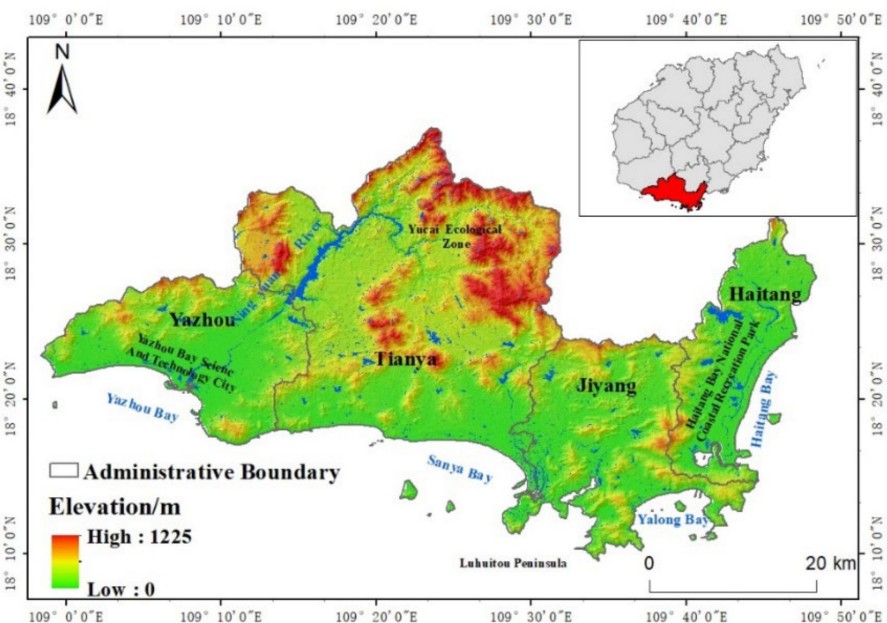

**Figure 1.** Location of Sanya City.

### 2.2. Data Sources and Processing

The data used in this study included remote sensing images and data for the driving factors of the land use simulation. The remote sensing images were selected from Landsat data, with a low degree of cloudiness and a spatial resolution of 30 m. The images were acquired in June 2010, January 2014, and February 2018. The data were obtained from https://www.gscloud.cn/, accessed on 24 October 2020. The image data were radiometrically calibrated and atmospherically corrected to eliminate radiation errors.

Combining the actual situation of Sanya and the Classification of Current Land Use standard (GB/T 21010-2017), the land use in Sanya was classified as cultivated land, forest, grassland, construction land, water, and unused land. The remote sensing image interpretation was performed using a supervised classification support vector machine algorithm to obtain three phases of land use classification data. Based on high-resolution Google Earth images, sample points were randomly selected to verify the classification results, and the average accuracy of the classification reached 86.25%, which meets the accuracy requirements.

In addition, a total of eight variables, including natural factors, transportation location factors, and socio-economic factors of the city, were selected as the driving factors of the land-use change simulation in Sanya conducted using the PLUS model. Among them, the natural factors included elevation and slope, and the elevation data were obtained from 30 m resolution digital elevation model (DEM) data, while the slope data were generated from the DEM data using the slope function spatial analysis tool in ArcGIS. The transportation location factor reflects the accessibility of each type, including the distance to rivers, railways, general roads, and town centers, and it was calculated and generated using the European distance tool in ArcGIS. Socio-economic factors are one of the important drivers of spatial development changes in a city. In this study, spatial raster data were generated based on the resident population and gross domestic product (GDP) of each administrative region in the Sanya Statistical Yearbook (http://tjj.sanya.gov.cn). The specific data sources are listed in Table 1.

**Table 1.** Data sources.

| | Name | Description | Source |
|---|---|---|---|
| | Landsat data | Land use cover/change (30 m) | Geospatial Data Cloud (https://www.gscloud.cn/, (accessed on 24 October 2020)) |
| Driving factors | Natural | DEM (30 m)<br>Slope (30 m) | National Geomatics Center of China (https://www.ngcc.cn/ngcc/, (accessed on 21 June 2021)) |
| | Transportation and location | Distance to river<br>Distance to rails<br>Distance to highway<br>Distance to centers in the city | Resource and Environment Science and Data Center (https://www.resdc.cn/, (accessed on 25 June 2021))<br><br>Euclidean Distance |
| | Socio-economic | Population density<br>GDP | Statistical Yearbook of Sanya City |

*2.3. Methods*

2.3.1. Remote Sensing Ecological Index

The RSEI is an integrated index that includes four factors: greenness, humidity, heat, and dryness, which reflect the vegetation index, the humidity of the vegetation and soil, the surface temperature, and the index of buildings and bare ground, respectively [31,32]. The remote sensing index is defined as

$$RSEI = f(NDVI, Wet, NDBSI, LST) \tag{1}$$

where NDVI is the greenness index, Wet is the humidity index, NDBSI is the dryness index, and LST is the heat index.

The humidity component of the tassel cap transformation was selected to represent Wet, which can reflect the humidity content of the vegetation, soil, and artificial ground surfaces in the study area. Due to the presence of large water bodies in the study area, the normalized difference water index was used to mask the water body information constructed by Mcfeeters [33] to keep the humidity indicator from being affected by the water bodies. In this study, Landsat8 operational land imager (OLI) imagery was used as an example, and the following equation was used to calculate the humidity index. Using Landsat8 OLI imagery as an example, the following equation was used to calculate the humidity indicator:

$$Wet = 0.1511B1 + 0.1973B2 + 0.3283B3 + 0.3407B4 - 0.7117B5 - 0.4559B6 \tag{2}$$

where B1 to B6 represent the blue, green, red, near-infrared, short-wavelength infrared (SWIR1), and SWIR2 bands of the Landsat8 OLI, respectively.

The normalized difference vegetation index (NDVI) is closely related to the plant biomass, leaf area index, and vegetation cover. It is the preferred indicator for analyzing vegetation changes and has been widely used in remote sensing monitoring of vegetation. Therefore, the NDVI was chosen to represent the greenness indicator in this study.

$$NDVI = \frac{B4 - B3}{B4 + B3} \tag{3}$$

The normalized difference building and soil index (NDBSI) represents the dryness index in the RSEI model, and the bare earth and building surfaces in urban areas are always in a dry state. Therefore, the NDBSI can be calculated in combination with the index-based built-up index (IBI) and the bare earth index (SI):

$$NDBSI = \frac{SI + IBI}{2} \tag{4}$$

$$IBI = \left[\frac{2B5}{B4 + B5} - \left(\frac{B4}{B3 + B4} + \frac{B2}{B2 + B5}\right)\right] / \left[\frac{2B5}{B4 + B5} + \left(\frac{B4}{B3 + B4} + \frac{B2}{B2 + B5}\right)\right] \tag{5}$$

$$SI = \frac{(B3 + B5) - (B4 + B1)}{(B3 + B5) + (B4 + B1)} \tag{6}$$

where SI is the bare earth index, and IBI is the building index.

The land surface temperature (LST) was used to represent the heat index, and it could be calculated using the single-channel algorithm proposed by Jimenez-Munoz et al. [34]:

$$LST = \gamma\left[\varepsilon^{-1}(\psi_1 L_{sen} + \psi_2) + \psi_3\right] + \delta \tag{7}$$

where $\varepsilon$ is the surface emissivity, and $\gamma$ and $\delta$ are two parameters of the Planck function. $L_{sen}$, $\psi_1$, $\psi_2$, and $\psi_3$ are the parameters related to the sensor and atmospheric parameters.

Finally, principal component analysis (PCA) was conducted to determine the relative importance of each variable, and the weights of each factor were automatically and objectively assigned according to the contribution of each factor to the principal component, which makes the results of the ecological quality evaluation more objective. Before the PCA was conducted, the four factors were normalized to the range of 0 to 1. The results of the PCA revealed (Table 2) that the first principal component (PC1) of the data in both periods contributed more than 85% to the RSEI, indicating that the PC1 contained most of the characteristics of the four indicators. Among the four indicators in PC1, the NDVI and Wet were positive, and the NDBSI and LST were negative, indicating that the greenness and humidity had a positive effect on the ecological environment of the study area, while the urban factors and temperature had a negative effect on the ecological environment. The highest load value of the NDVI in PC1 indicates that it had the greatest impact on the ecological quality. The RSEI values were normalized to values of 0–1 again using the first principal component, with values closer to 1 representing a better ecological quality.

**Table 2.** Principal component analysis (PCA) statistical.

| Year | Index | PC1 | PC2 |
|------|-------|-----|-----|
| 2014 | NDVI | 0.803 | 0.567 |
| | Wet | 0.229 | −0.478 |
| | NDBSI | −0.523 | 0.526 |
| | LST | −0.170 | 0.416 |
| | Eigenvalue | 0.0794 | 0.0080 |
| | Contribution rate/% | 87.82 | 8.83 |
| 2018 | NDVI | 0.774 | 0.603 |
| | Wet | 0.241 | −0.432 |
| | NDBSI | −0.501 | 0.398 |
| | LST | −0.303 | 0.539 |
| | Eigenvalue | 0.0781 | 0.0076 |
| | Contribution rate/% | 86.40 | 8.45 |

### 2.3.2. Patch-generating Land Use Simulation Model

The PLUS model is mainly trained by extracting random samples from land change data using a random forest algorithm, digging deeply into the driving and expansion factors of the land-use change, and finally outputting the development probability of the land use type in the cell [29]:

$$P_{i,k}^d(x) = \frac{\sum_{n=1}^{M} I(h_n(x) = d)}{M} \tag{8}$$

where the value of d is 0 or 1 (1 indicates the conversion of other land-use types to land use type k, and 0 indicates other conversions), x is a vector composed of multiple drivers, $I(\cdot)$ is an indicator function of the decision tree set, $h_n(x)$ is the prediction type of the nth decision tree of vector x, and M is the total number of decision trees.

The PLUS model uses a multi-type random patch seeding mechanism based on the threshold drop to predict the patch evolution. This mechanism generates a change seed on

the development probability plane of each land-use type using the Monte Carlo method when the neighborhood effects of land use type k are 0. The expression is

$$OP_{i,k}^{1,t} = \begin{cases} P_{i,k}^1 \times (r \times \mu_k) \times D_k^t & \text{if } \Omega_{i,k}^t = 0 \text{ and } r < P_{i,k}^1 \\ P_{i,k}^1 \times \Omega_{i,k}^t \times D_k^t & \text{all others} \end{cases} \tag{9}$$

where r is a random value in the range of 0 to 1, and $\mu_k$ is the threshold value for generating new land-use patches, $P_{i,k}^1$ is the growth probability of land-use type k in cell i, and $\Omega_{i,k}^t$ is the neighborhood effect of cell i. $D_k^t$ is the adaptive inertia coefficient, which depends on the difference between the current land use quantity and the target demand quantity at iteration t and indicates the impact of future land demand on land use type k.

In this study, the samples were extracted using the uniform sampling method, by combining the eight driving factors, employing the random forest classification algorithm, and generating the growth probability of the Sanya land use classes. Second, the model requires input conditions such as the adaptive inertia coefficients, neighborhood parameters, and a transformation matrix. The Markov model was used to simulate the land-use change demand in the future based on the amount of land use in both periods, which was used to calculate the adaptive inertia coefficients in the model. The neighborhood effect describes the expansion intensity of the land-use class, which is the ability of each land-use class to expand by itself, driven by external factors. The threshold value ranges from 0 to 1, and the closer the value of the neighborhood weight parameter is to 1, the stronger the expansion ability is. The conversion cost matrix describes the possibility of converting from an initial type to a demand type. The matrix is set to 0 when one type is not allowed to be converted to another; otherwise, it is set to 1. The conversion cost matrix describes the possibility of converting from the initial type to the demand type. The matrix value is set to 0 when one type of land is not allowed to be converted to another; otherwise, it is set to 1. The above values were derived from the literature. Finally, by combining the growth probability of the land use classes, adaptive inertia coefficients, neighborhood weights, and conversion cost matrix, the simulation results of the urban built-up growth in 2018 were obtained. The results were verified through comparison with the actual land use classification data for 2018. The urban built-up growth of Sanya in 2030 was simulated using the same method.

After the urban growth simulation, due to the existence of a large number of scattered and small-area patches of built-up land, after these patches were removed, the UGB was delineated on this basis.

## 3. Results and Analysis

### 3.1. Ecological Quality Evaluation of Sanya

From 2014 to 2018, the mean RSEI in Sanya decreased from 0.656 to 0.632, indicating a decrease in the overall ecological quality (Table 3). The NDVI and Wet, which had a positive impact on the ecological environment of Sanya, increased from 0.744 to 0.749 and from 0.587 to 0.603, respectively. For the remaining two indicators that negatively affected the ecological environment, the mean value of the LST exhibited a decreasing trend, while the mean value of the NDBSI increased. Although the increases in the NDVI and Wet and the decrease in the LST indicated an improvement in the ecological environment, the increase in the NDBSI made a greater contribution to the ecological damage than the first three, resulting in a decrease in the overall RSEI. This verifies that the ecological quality of Sanya was damaged by the increase in the artificial buildings and bare land due to urban development and construction.

**Table 3.** The values of ecological indicators and RSEI.

| Year | NDVI | Wet | NDBSI | LST | RSEI |
|------|------|------|-------|------|------|
| 2014 | 0.744 | 0.587 | 0.514 | 0.537 | 0.656 |
| 2018 | 0.749 | 0.603 | 0.526 | 0.391 | 0.632 |

The RSEI results were divided into five categories according to the RSEI values using equal intervals: excellent (0.8–1.0), good (0.6–0.8), medium (0.4–0.6), poor (0.2–0.4), and very poor (0.0–0.2) [32]. As can be seen from Figure 2, the areas with excellent ecological quality in Sanya were mainly located in the mountains in the northern part of the Tianya District and on the Luhuitou Peninsula in the south. The mountain ranges bordering the Haitang and Jiyang districts had a high degree of forest coverage and were less affected by human activities. The areas with poor ecological quality were mainly located in the built-up areas in Yazhou Bay town and in the central part of the city, as well as in the Haitang Bay National Coastal Recreation Park. The percentages of areas with excellent, good, and medium ecological environment qualities in both periods were >87%, indicating that Sanya had a good ecological environment background (Table 4).

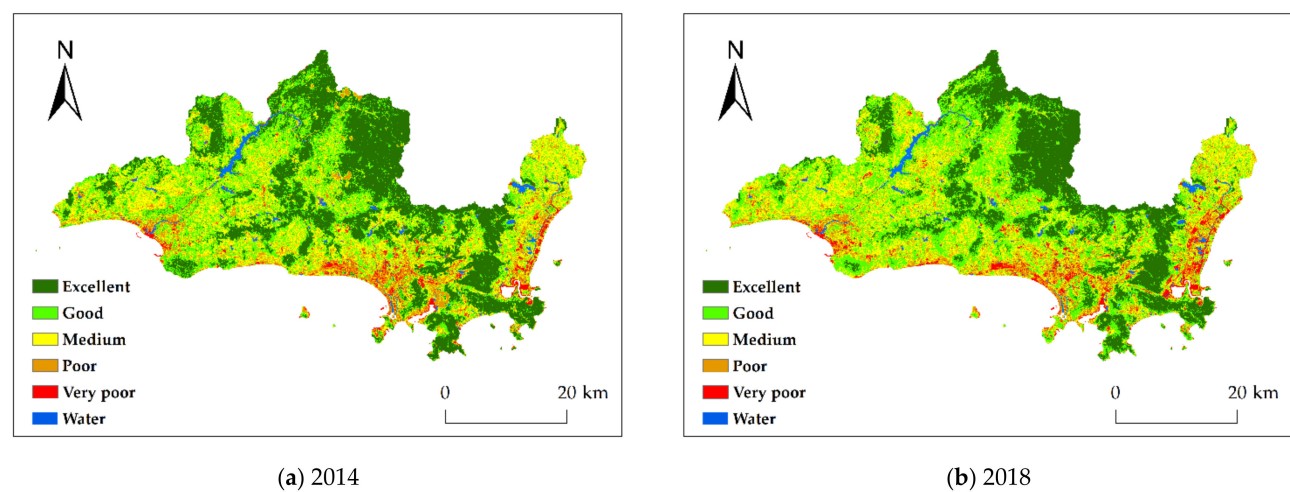

(**a**) 2014                                           (**b**) 2018

**Figure 2.** RSEI of Sanya in 2014 and 2018.

**Table 4.** Statistics of RSEI grades in 2014 and 2018.

| Grade of RSEI | 2014 RSEI | | 2018 RSEI | | Changes in 2104–2018 | |
|---|---|---|---|---|---|---|
| | Area/km$^2$ | Percentage/% | Area/km$^2$ | Percentage/% | Area/km$^2$ | Percentage/% |
| 1: Poor (0.0–0.2) | 32.8 | 1.79 | 47.3 | 2.58 | 14.5 | 44.16 |
| 2: Very poor (0.2–0.4) | 179.0 | 9.76 | 187.4 | 10.22 | 8.4 | 4.70 |
| 3: Medium (0.4–0.6) | 475.3 | 25.93 | 501.5 | 27.36 | 26.2 | 5.52 |
| 4: Good (0.6–0.8) | 534.0 | 29.13 | 617.1 | 33.66 | 83.1 | 15.56 |
| 5: Excellent (0.8–1.0) | 611.9 | 33.38 | 479.8 | 26.17 | −132.2 | −21.60 |

From 2014 to 2018, the excellent RSEI grade area decreased by 132.2 km$^2$, mostly changing to the good and medium grades. The area of poor and very poor ecological quality increased by a total of 22.9 km$^2$, so the overall ecological quality of Sanya deteriorated. The areas with an obvious decline in ecological quality were the coastal areas of the Haitang District and the Ningyuan River Basin in the Yazhou District.

Based on the RSEI results, it was found that due to the construction of the Science and Technology Industrial Park in the Yazhou District and the development of hotels and real estate driven by the tourism development along the Haitang Bay National Coast, the construction land area in these two areas increased significantly, leading to decreases in the regional NDVI and Wet and increases in the LST and NDBSI. This was the main reason for the decrease in the overall RSEI in Sanya (Figure 2). The mountainous and ecological conservation areas in the northern part of Sanya, on the Luhuitou Peninsula in the south, and in the mountain ranges at the junction of the Haitang and Jiyang districts had relatively high elevations and were mainly covered by forest. The four indicators were relatively

stable in these areas, and these areas played an important supporting role in maintaining the stability of the overall RSEI in Sanya.

The excellent and good RSEI grade areas were mainly distributed in the northern mountains in the Tianya District and the mountain range at the junction between the Jiyang and Haitang districts. These regions contained mountain, forest, lake, and river ecological protection areas, covering the tropical ecological park, mountain and sea corridor, coastal mangrove wetlands, river systems, and other ecologically fragile areas in the key ecological zones in the territorial spatial planning for Sanya. These areas were extremely sensitive to the effects of human activities, and in the future, they should be strictly controlled as ecological protection zones for urban construction and development.

### 3.2. Urban Built-Up Land Expansion Simulation

The urban spatial pattern of Sanya exhibited a strip-shaped distribution, and the built-up area was mainly distributed in the central part of the city (including the Tianya and Jiyang districts), the Yazhou District, and the Haitang District (Figure 3). Under the policy background of building an international tourism island in 2010, the urbanization process of Sanya has accelerated. In 2014, the area of Sanya's built-up land area was 122.1 km$^2$, accounting for 6.66% of the city. In 2018, the built-up land area was 171.5 km$^2$, accounting for 9.35%. The built-up land area of Sanya increased by 49.4 km$^2$ from 2014 to 2018, with an increase of 40.4% and an annual increase of 12.3 km$^2$. The trend of urban spatial growth was coastal to inland expansion, and the three regions exhibited a trend of integration and connectivity. Based on the statistical data for Sanya, in 2014, the GDP of Sanya was 40.438 billion yuan, and the population was 585,600. In 2018, the GDP was 59.551 billion yuan, and the population was 614,600. The GDP growth of Sanya from 2014 to 2018 was 47.26%, and the population growth was 5%. The high economic development and population growth were the main reasons for the expansion of the urban area in Sanya from 2014 to 2018.

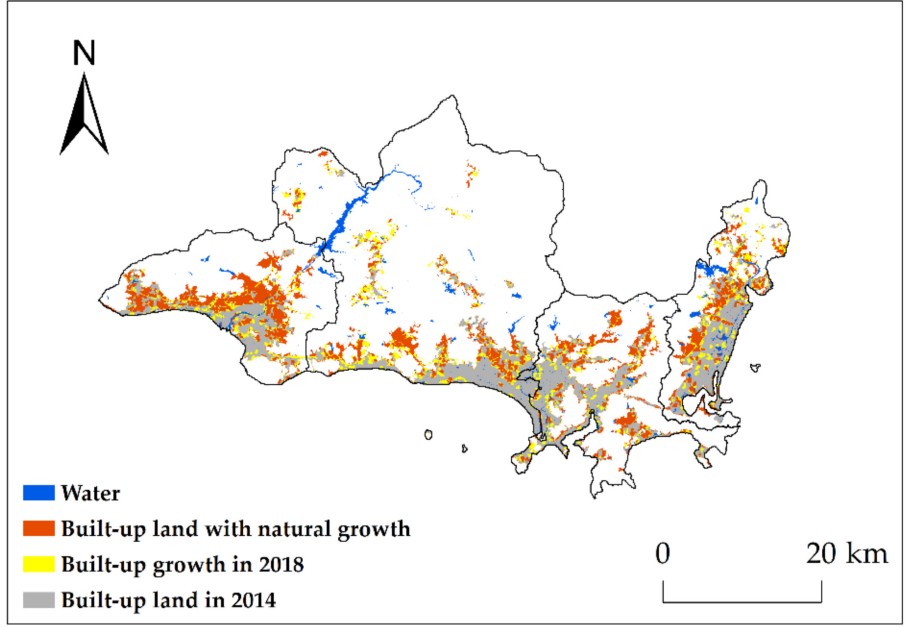

**Figure 3.** Urban built-up land expansion trend.

The simulation of the UGB of Sanya under the natural growth scenario was based on the land-use change trend from 2014 to 2018 and the PLUS model. The results show that the kappa coefficient and overall accuracy of the simulation results are 0.841 and 89.28%, respectively, indicating that the accuracy of the model simulation is high. Thus, it is suitable for simulating the growth of the urban built-up area in Sanya in 2030. The built-up area in Sanya in 2030 under the natural development scenario increases significantly, and the

built-up land area reaches 298.0 km$^2$, accounting for 16.26%, which is 126.6 km$^2$ higher than that in 2018. Sanya's urban spatial growth mainly occurs in the central part of the city, the Yazhou District, and the Haitang District. The growth of the urban space in the Yazhou District is dominated by expansion in two directions: along the coast of Yazhou Bay, and in the downstream area of the Ningyuan River. The land use classification data show that there is a large amount of cultivated land in this area, so the urban growth occupies the agricultural space. Second, in the area along the coastline of Haitang Bay, the overall building of the region is denser, and the town expansion exhibits a trend of dispersal. The third area is the central part of Sanya, which exhibits a more obvious trend of spreading from the central mountainous area.

Under the natural growth scenario, the urban built-up area occupies an area of approximately 112.8 km$^2$ that previously had excellent and good ecological qualities, with an area proportion of 10.28% of the bottom-line ecological pattern of Sanya, which poses a serious threat to ecological security. Therefore, to avoid damaging the ecology and agriculture due to the disorderly expansion of the city, the actual situation of Sanya should be considered, and the bottom-line ecological security pattern determined by the RSEI evaluation should be used as rigid boundaries of the urban development to prohibit urban development and occupation.

### 3.3. UGB Integrated Ecological Constraint

In this study, the excellent and good RSEI areas were used to construct the Sanya ecological constraint zone, and the PLUS model was used to simulate the urban built-up growth in 2030 and to delineate the UGB (Figure 4). The results show that under the ecological constraints, the built-up area in 2030 is 218.4 km$^2$, accounting for 11.92% of the city, and the built-up land is 79.6 km$^2$ less than under the natural growth scenario (Figure 5). The urban built-up growth in 2030 based on the ecological constraints is different from the sprawl trend under the natural scenario. It is characterized by connotative growth, and the main growth space is within the built-up area of the city. Since the industrial development layout of Sanya focuses on key areas, such as the Yazhou Bay Science and Technology City, the Sanya Central Business District, and the Haitang Bay National Coastal Leisure Park, the key industrial development areas are the central part of the city, Haitang Bay, and the Yazhou District. The built-up areas still expand, and these areas are the focus of Sanya's urban development in the future.

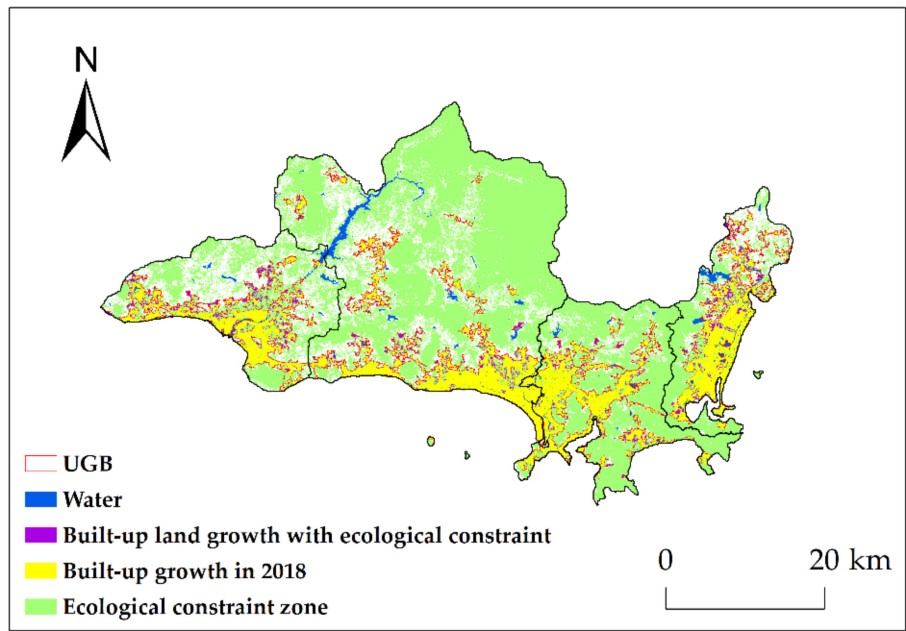

**Figure 4.** Urban built-up growth simulation and UGB delineation in 2030.

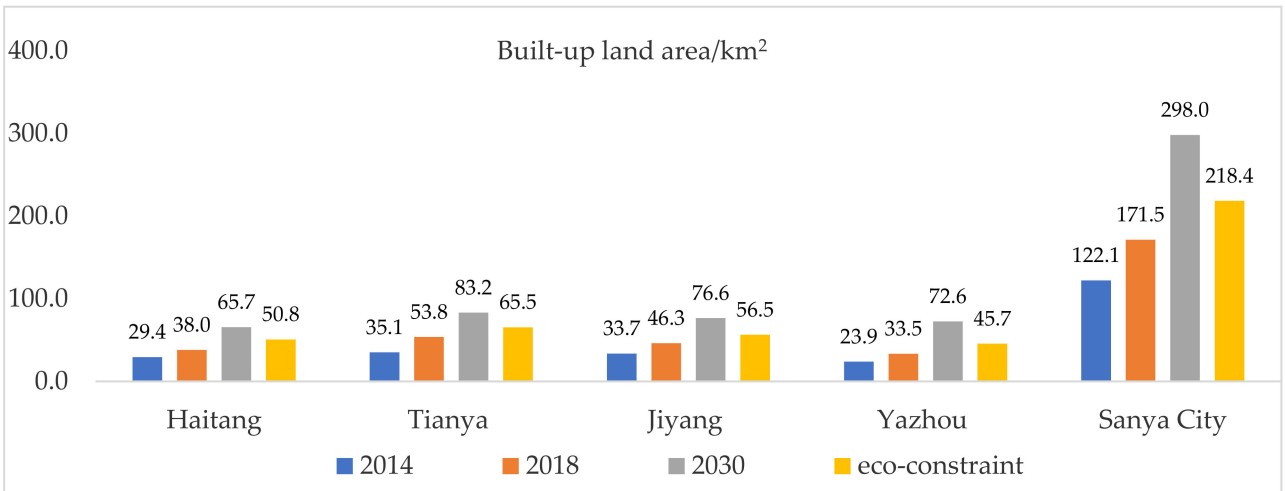

**Figure 5.** Built-up land area in different years.

The central part of the city is mainly characterized by connotative growth, and the built-up land in the Yazhou and Haitang districts exhibits a connotative and epitaxial growth trend, with growth rates of 21.80%, 36.52%, and 33.73%, respectively, in these three regions compared with 2018.

Compared with the growth under the natural scenario, the UGB delineation coupled with the RSEI not only effectively restrains the blind expansion of the built-up land but also effectively protects the ecological and agricultural security. In addition, there is still room for urban development in the northwestern part of the Yazhou District, which reserves space resources for future urban industrial layouts.

## 4. Discussion

Urban sprawl encroaches on ecological spaces, which is not conducive to the economic and intensive use of land and causes waste of land resources. In addition, an increase in the urban population will also bring about an increase in production and living pollution, which will put more pressure on the environment. Third, urban sprawl will destroy the valuable tropical rainforest resources in the tropics, and the loss of vegetation and shallow soil will aggravate the problem of soil erosion in the tropics. To maintain ecological security, China has successively formulated a series of laws and regulations, such as the Environmental Protection Law, Marine Environmental Protection Law, Land Management Law, Wildlife Protection Law, Forest Law, Grassland Law, Water Law, Water and Soil Conservation Law, and Regulations on the Management of Nature Reserves, in order to promote ecological protection and ecological construction in China.

Sanya is an important tropical coastal tourist city in China. With the booming tourism industry, accelerating urbanization, and increasing population, Sanya's natural resources and ecological environment are being put under increasing pressure. Most of Sanya's tourism is based on natural ecological landscapes, and the construction of tourism facilities and excessive commercial development in these areas have exceeded the environmental carrying capacity of the ecological area, which is not conducive to the protection of natural resources and habitats. Second, Sanya's unique tropical natural resources provide significant advantages in tropical agricultural cultivation. The local government encourages local farmers to vigorously cultivate tropical crops, and tropical cultivation has brought great social benefits to the region. However, tropical plantations mainly grow single tropical forest fruits, such as mangos, bananas, and betel nuts. Since most of the plantations are located in hilly and mountainous areas, there is a lack of herbaceous vegetation under the cultivated forests. This tends to cause serious soil erosion and biodiversity destruction, which is not conducive to sustainable rural land and farmer development. Third, regarding the government, real estate development investment has become the main driver of

economic development in Sanya, and ecological spaces can only be sacrificed if spatial resources are increasingly saturated and the red line of arable land exists [35]. These are the main factors preventing the protection of the ecological areas in Sanya.

According to the Regulations for the Administration of Ecological Protection Red Lines in Hainan Province, the government entities at or above the county level can demarcate the strictly ecological control boundaries in areas such as land and marine key ecological functional areas, eco-environmentally sensitive areas, and fragile areas within their administrative region. The ecological constraint area of Sanya based on the RSEI basically realizes this ecological boundary delineation, and it is consistent with the ecological conservation pattern in the Sanya Territory Spatial Planning (2020–2035) (http://zgj.sanya.gov.cn/, (accessed on 20 November 2021)), indicating that the RSEI is highly scientific and rational. The RSEI is completely based on remote sensing data and a variety of ecological factors contained in the index, so it can be used to comprehensively evaluate the ecosystem and to monitor the ecological quality promptly. In addition, the RSEI is an ideal way to monitor the quality of the urban environment in tropical areas due to its accessibility, objectivity, and timeliness. Urbanization has inhibited the improvement of ecological quality and has generally damaged the ecological environment. In recent years, the urban expansion in Sanya has inevitably occupied natural ecological resources such as forests, agricultural land, wetlands, and shorelines, forming impervious water surfaces. In addition, many bare surfaces are formed during the construction process. Studies have shown that the impervious surfaces of buildings and the increase in the surface of the Earth have significantly aggravated the thermal environment of the region, further triggering the urban heat island effect, destroying the regional ecological balance, and seriously threatening biodiversity [36]. The heat island effect amplifies the actual effects of summer urban heat waves, causing serious thermal hazards. This suggests that the ecological damage caused by urban sprawl is particularly significant in tropical cities, and the results of this study also verify this. Based on the above analysis, cities in tropical areas should pay attention to the construction of ecological cities, slope monitoring, bare slope treatment, vegetation ecological restoration, and soil and water conservation in the construction process. They should also maintain ecological security and effectively reduce the ecological and environmental problems caused by urban expansion.

Urban land expansion is a complex process, and UGB delineation needs to consider the comprehensive effects of multiple factors. In this study, the ecological constraint concept of anti-planning and a complex system simulation model were used to solve this problem. The PLUS model can handle complex land-use changes and make the simulation results more consistent with the real planning; therefore, the PLUS model has a good potential for application in urban built-up growth simulations. Under both scenarios, the area of future urban built-up land increases, but the magnitudes and directions of the increases are very different. The results for the natural growth scenario show that in 2030, the urban built-up expansion will occupy a large amount of agricultural and ecological space, sacrificing cultivation and ecological security. The scenario that integrates the RSEI ecological constraints negates the extension of the urban sprawl that occurs under the natural growth scenario and emphasizes the importance of connotative growth and stock redevelopment under the premise of protecting the ecological space.

Because there is no previous research on the boundary delineation of Sanya, we compared the delineation results with the Sanya Territory Spatial Planning (2020–2035), and it was found that the results are more consistent with the planning. The northern mountainous area of Sanya is an important core ecological conservation area, which is planned as a tropical ecological park and an ecological mountain area integrating rural landscapes. It is also a boundary of the urban development constraints identified in this study. In addition, the basic idea of Sanya's urban planning is to establish three major urban areas that integrate industry and city and balance work, housing, and services. That is, an international tourism and consumption center will be built in the Haitang Bay area. In the Yazhou Bay area, high-tech industries will gather to build deep-sea technology and

Nanfan technology city. As a regional comprehensive service center, the central urban area will mainly strengthen the regional radiation capability, integrate existing space resources, and optimize the facilities and the image of the city center in the future. The urban growth simulation for these areas meets the planning requirements, and industrial space is reserved for future development in the northwestern region. Therefore, under the framework of the current policy-led territorial spatial planning and management, the ecologically constrained UGB simulation method can more realistically reflect the future urban spatial layout and structure and can effectively guide the planning layout. To effectively play the role of UGB, it is necessary to emphasize the constraints of the UGB on urban growth and protect the ecological environment by inhibiting the disorderly expansion of the city in the implementation process. Moreover, there should be flexibility to support the rational development of the city. The UGB is not static, and the UGB delineation based on the urban growth space needs to be dynamically adjusted according to the changes in the ecological environment and urban development needs.

There are still some shortcomings of this study that need to be further explored. First, climate change is considered to be one of the important natural factors affecting ecosystems, and its ecological impact has many uncertainties [37]. In this study, the impacts of climate factors on the urban ecological quality were not considered, and thus, future research needs to incorporate these factors. Second, urban development simulation is a complex process. In addition to considering the macro policy factors of planning, it also needs to consider the bottom-up growth and the impacts of different agents on urban development, that is, quantitative analysis of the trade-offs between different agents. In future research, the urban spatial development simulation will be optimized through the use of a combination of macro and micro factors.

## 5. Conclusions

From the perspective of the sustainable development of tropical cities, based on ecological quality assessment, a new method of simulating the spatial growth of tropical cities based on ecological constraints was developed in this study.

According to the evaluation results obtained using the RSEI, the ecological quality background of Sanya is relatively good, but it decreased significantly from 2014 to 2018, which was mainly due to the development and construction of the key development areas in Sanya. Based on the urban spatial development trend from 2014 to 2018, the growth of the urban built-up land in 2030 was simulated. The results show that this scenario leads to the extensive occupation of ecological and agricultural space and is unsustainable. It is necessary to establish an RSEI-based ecological bottom-line pattern as a rigid boundary for urban development.

Simulation of future urban expansion areas and UGB delineation based on the RSEI-PLUS model can better balance the conflict between ecological protection and urban expansion. Under the premise of ecological land protection, the urban space can also be rationally configured according to the local conditions, and it also accurately reflects the urban spatial expansion under the background of territory spatial planning. Therefore, the RSEI-PLUS-based approach to simulating urban spatial growth is an innovative, highly feasible approach that can effectively support the current territorial spatial planning.

Due to the uniqueness of natural resources in tropical areas, regional cities are facing problems in the sustainable development of the current economic development, population growth, and urban expansion, as well as the environmental carrying and protection. The results of this study provide solutions for the socio-economic and ecologically sustainable development of tropical cities.

**Author Contributions:** Conceptualization, N.H. and P.J.; methodology, N.H.; data curation, K.H., M.Y. and Y.Z.; writing—original draft, N.H. and K.H.; writing—review and editing, N.H., K.H. and P.J.; funding acquisition, P.J. All authors have read and agreed to the published version of the manuscript.

**Funding:** This research was funded by the National Social Science Foundation of China, No. 21XGL019, and the Hainan Provincial Natural Science Foundation of China, No. 421RC1034.

**Institutional Review Board Statement:** Not applicable.

**Informed Consent Statement:** Not applicable.

**Data Availability Statement:** The data presented in this study are openly available in Geospatial Data Cloud (https://www.gscloud.cn/, (accessed on 24 October 2020)), National Geomatics Center of China (https://www.ngcc.cn/ngcc/, (accessed on 21 June 2021)), Resource and Environmental Science and Data Center (https://www.resdc.cn/, (accessed on 25 June 2021)) and Statistical Yearbook of Sanya City.

**Acknowledgments:** The authors gratefully acknowledge the two anonymous reviewers for their valuable comments and suggestions, which strengthened the quality of the paper substantially.

**Conflicts of Interest:** The authors declare no conflict of interest.

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
