# Peer review of "Incorporating Ecological Constraints into the Simulations of Tropical Urban Growth Boundaries: A Case Study of Sanya City on Hainan Island, China"

_applsci, doi:10.3390/app12136409_

Round 1
Reviewer 1 Report
Thank you for an interesting paper. Below please find comments and suggestions, large and small:
- This paper really needs to be edited by a native English speaker.
- When referencing U.S. cities, it is imperative to include the state for places that are not major cities because there is more than one Salem and more than one Portland.
- Please mention on first reference in the title, abstract and paper that Sanya City is in China.
- Please discuss whether or not there have been previous such studies on Sanya City and if so what they studies showed and how the results compare to your study.
- In Figure 2, it is unclear what is being shown — excellent, good, middle, etc., are adjectives, so what are they describing?
- In Figures 3 and 4, the yellow on the map and the yellow in the key do not appear to match.
- In Figure 5, please adjust the numbers so that they do not overlap each other and please capitalize the "c" in Sanya City.
- Please discuss any laws that either encourage or prevent protect of the threatened ecological areas.
- Please discuss the costs of expansion into the ecological areas and the social, economic and political barriers preventing the protection of the areas.
- Please describe what agencies if any have jurisdiction to decide what areas are protected.
Author Response
Dear Reviewer,
Thank you for your comments on our manuscript entitled “applsci-1733375”, These comments are of great value to the improvement of our research. According to the comments on the manuscript review, the responses are as follows:
Point 1: This paper really needs to be edited by a native English speaker.
Response 1: This paper has been polished by a native English speaker.
Point 2: When referencing U.S. cities, it is imperative to include the state for places that are not major cities because there is more than one Salem and more than one Portland.
Response 2: Salem and Portland are both located in Oregon, USA, and Oregon has been added to the article.
Point 3: Please mention on first reference in the title, abstract and paper that Sanya City is in China.
Response 3: It has been mentioned in the first reference of the title, abstract and paper that Sanya is located in China.
Point 4: Please discuss whether or not there have been previous such studies on Sanya City and if so what they studies showed and how the results compare to your study.
Response 4: Because there is no previous research on the boundary delineation of Sanya, we compared the delineation results with the Sanya Territory Spatial Planning (2020–2035), and it was found that the results are more consistent with the planning.The above has been added in lines: 469-471.
Point 5: In Figure 2, it is unclear what is being shown — excellent, good, middle, etc., are adjectives, so what are they describing?
Response 5: Each of these adjectives indicates the categories of RSEI value, which is described in the section3.1: “The RSEI results were divided into five categories according to the RSEI values using equal intervals, namely, excellent (0.8-1.0), good (0.6-0.8), middle (0.4-0.6), poor (0.2-0.4), and very poor (0.0-0.2)”.
Point 6: In Figures 3 and 4, the yellow on the map and the yellow in the key do not appear to match.
Response 6: The ranges indicated by the yellow legend in Figure 3 and Figure 4 is not the same. The yellow legend in Figure 3 represents the built-up growth in 2018, and the total built-up land in 2018 is the built-up growth in 2018 plus the built-up land in 2014. The yellow legend in Figure 4 refers to the existing built-up land in 2018.
Point 7: In Figure 5, please adjust the numbers so that they do not overlap each other and please capitalize the "c" in Sanya City.
Response 7: Figure 5 have been adjusted, the numbers do not overlap each other and delete the "city" in Sanya City.
Point 8: Please discuss any laws that either encourage or prevent protect of the threatened ecological areas.
Response 8: The following has been added to the Discussion section in lines: 404-409:
To maintain ecological security, China has successively formulated a series of laws and regulations, such as the Environmental Protection Law, Marine Environmental Protection Law, Land Management Law, Wildlife Protection Law, Forest Law, Grassland Law, Water Law, Water and Soil Conservation Law, and Regulations on the Management of Nature Reserves, in order to promote ecological protection and ecological construction in China.
Point 9: Please discuss the costs of expansion into the ecological areas and the social, economic and political barriers preventing the protection of the areas.
Response 9: The following has been added to the Discussion section in lines: 399-404 and 410-427.
Urban sprawl encroaches on ecological spaces, which is not conducive to the economical and intensive use of land and causes waste of land resources. In addition, an increase in the urban population will also bring about an increase in production and living pollution, which will put more pressure on the environment. Third, urban sprawl will destroy the valuable tropical rainforest resources in the tropics, and the loss of vegetation and shallow soil will aggravate the problem of soil erosion in the tropics.
Sanya is an important tropical coastal tourist city in China. With the booming tourism industry, accelerating urbanization, and increasing population, Sanya's natural resources and ecological environment are being put under increasing pressure. Most of Sanya's tourism is based on natural ecological landscapes, and the construction of tourism facilities and excessive commercial development in these areas have exceeded the environmental carrying capacity of the ecological area, which is not conducive to the protection of natural resources and habitats. Second, Sanya's unique tropical natural resources provide significant advantages in tropical agricultural cultivation. The local government encourages local farmers to vigorously cultivate tropical crops, and tropical cultivation has brought great social benefits to the region. However, tropical plantations mainly grow single tropical forest fruits, such as mangos, bananas, and betel nuts. Since most of the plantations are located in hilly and mountainous areas, there is a lack of herbaceous vegetation under the cultivated forests. This tends to cause serious soil erosion and biodiversity destruction, which is not conducive to sustainable rural land and farmer development. Third, regarding the government, real estate development investment has become the main driver of economic development in Sanya, and ecological spaces can only be sacrificed if spatial resources are increasingly saturated and the red line of arable land exists (Du Q and Du D H, 2018). These are the main factors preventing the protection of the ecological areas in Sanya.
Point 10: Please describe what agencies if any have jurisdiction to decide what areas are protected.
Response 10: According to the Regulations for the Administration of Ecological Protection Red Lines in Hainan Province, the government entities at or above the county level can demarcate the strictly ecological control boundaries in areas such as land and marine key ecological functional areas, eco-environmentally sensitive areas, and fragile areas within their administrative region. The above has been added to the Discussion section in lines: 428-432.
Reference:
[1] Du Qun and DU Dian-hu. The Innovation of legal mechanisms for Urban Mountains Resources Conservation in Light of Xi Jinping’s Ecological Ideas of Ruling by Law-A Case Study of the Regulations of Sanya City for Mountains Conservation[J].JOURNAL OF BEIJING NORMAL UNIVERSITY(SOCIAL SCIENCES),2018(04):5-14.

Reviewer 2 Report
To my understanding, in this manuscript, the authors tried to define/delineate urban boundaries through integrating RSEI with PLUS modeling. The objectives, methods, results and discussion generally look good to me. Please see my detailed comments below:
Lines 1-3: I would suggest revising the title to "Incorporating ecological constraints into the simulations of tropical urban growth boundaries: A case study of Sanya City on Hainan Island, China"
Lines 15, 20 & 23: PLUS, UGB, RSEI -> their full names should be spelled at least once before acronyms are used afterwards
Line 31: the decline of ecosystem service functions -> the degradation of ecosystem services
Line 37: 1976s -> 1970s
Line 101: simulate -> simulating
Line 116: marine -> coastal ?
Line 129: I would suggest further clarifying the position of the study area, by either showing a bigger map of China with Hainan Island highlighted, or labeling the latitudes and longitudes of the map corners
Line 209: I guess "NDVI and Wet" should be replaced by urban and temperature?
Lines 256-258: This seems to be the validation of PLUS results. If so, I think they should be moved to the "Results and Analysis" section, where a separate subsection of model validation results are presented.
Lines 357-359: This sentence needs a subject
Author Response
Dear Reviewer,
Thank you for your comments on our manuscript entitled “applsci-1733375”, These comments are of great value to the improvement of our research. We have responded according to the comments on the manuscript review,please see the attachment.

Round 2
Reviewer 1 Report
Thank you for a much-improved manuscript. When you submit another version, please improve the article's literature review.
Author Response
Dear reviewer:
Thank you for your modification suggestion.
The article's literature review has been improved , and it is marked in blue in the introduction section of the article.
